# Water Deficit Timing Differentially Affects Physiological Responses of Grapevines Infected with *Lasiodiplodia theobromae*

**DOI:** 10.3390/plants11151961

**Published:** 2022-07-28

**Authors:** Lia-Tânia Dinis, Cláudia Jesus, Joana Amaral, Aurelio Gómez-Cadenas, Barbara Correia, Artur Alves, Glória Pinto

**Affiliations:** 1Department of Agronomy & CITAB–Centre for the Research and Technology of Agro-Environmental and Biological Sciences, University of Trás-os-Montes e Alto Douro (UTAD), Apt. 1013, 5000-801 Vila Real, Portugal; 2Centre for Environmental and Marine Studies (CESAM), Department of Biology, University of Aveiro, Campus Universitário de Santiago, 3810-193 Aveiro, Portugal; cmjesus@ua.pt (C.J.); jsamaral@live.ua.pt (J.A.); bscorreia@ua.pt (B.C.); artur.alves@ua.pt (A.A.); gpinto@ua.pt (G.P.); 3Department de Ciències Agràries i del Medi Natural, Universitat Jaume I, E-12071 Castellón de la Plana, Spain; aurelio.gomez@uji.es

**Keywords:** defense mechanisms, plant physiology, hormones, Botryosphaeria dieback, water deficit timing × pathogen interaction

## Abstract

Diseases and climate change are major factors limiting grape productivity and fruit marketability. *Lasiodiplodia theobromae* is a fungus of the family Botryosphaeriaceae that causes Botryosphaeria dieback of grapevine worldwide. Abiotic stress may change host vitality and impact susceptibility to the pathogen and/or change the pathogen’s life cycle. However, the interaction between both stress drivers is poorly understood for woody plants. We addressed the hypothesis that distinct morpho-physiological and biochemical responses are induced in grapevine (*Vitis vinifera*)–*L. theobromae* interactions depending on when water deficits are imposed. Grapevines were submitted to water deficit either before or after fungus inoculation. Water deficit led to the reduction of the net photosynthetic rate, stomatal conductance, and transpiration rate, and increased the abscisic acid concentration regardless of fungal inoculation. *L. theobromae* inoculation before water deficit reduced plant survival by 50% and resulted in the accumulation of jasmonic acid and reductions in malondialdehyde levels. Conversely, grapevines inoculated after water deficit showed an increase in proline and malondialdehyde content and all plants survived. Overall, grapevines responded differently to the primary stress encountered, with consequences in their physiological responses. This study reinforces the importance of exploring the complex water deficit timing × disease interaction and the underlying physiological responses involved in grapevine performance.

## 1. Introduction

*Vitis vinifera* L. is a very important commercial fruit crop worldwide, covering a global area of approximately 7.5 million hectares. Besides grapevine berries being mainly used for wine production, grapevine is also exploited for the production of fresh table grapes, dried fruit (raisins), juice, tannins, and antioxidants [1]. Grapevines are adapted to continuously deal with a plethora of biotic and abiotic stresses [2]. However, this delicate adaptive balance has been threatened due to shifts in global climate patterns related to temperature increases and severe water deficit periods that may favour the incidence of plant diseases [3], changing host vulnerability and/or pathogen behavior. Grapevine trunk diseases (GTDs) are among the most destructive fungal diseases of *Vitis vinifera* worldwide, as there are still no effective control measures available [1,4]. These diseases represent a true challenge for viticulture, affecting the vineyard heritage and causing serious economic losses in this industry [1].

Botryosphaeria dieback, one of the main GTDs, is caused by several species of Botryosphaeriaceae, including *Lasiodiplodia theobromae* [1,4]. Common signs of *L. theobromae* infection in grapevine include leaf spots and wilting, bud necrosis or mortality, dead arms, shoot dieback, and bunch rot [1,4,5]. Furthermore, infections may lead to the development of brown stripes below the bark and wedge-shaped necrotic sectors [1,4,5]. *Lasiodiplodia theobromae* is widely distributed, but is more prevalent in tropical and subtropical regions e.g., [5]. This pathogenic fungus is recognized as latent in many woody plants, including grapevines [1], being able to survive endophytically in their hosts and transmit into a pathogenic phase after the onset of abiotic stress, including high temperatures and drought [4,6,7,8].

Grapevines are well-adapted to moderate water deficits, which can even enhance wine quality if occurring during maturation [9]. However, the occurrence of frequent drought periods—especially in regions with a Mediterranean climate—impairs overall grapevine performance, with negative impacts on grape yield and quality [10]. Plant responses to water deficits are complex and dependent on the duration and intensity of the stress [11]. The regulation of stomata closure by abscisic acid is a key grapevine response to prevent water losses and to protect against more severe damage such as leaf cavitation and shedding during water deficits [12,13], also resulting in reductions in carbon assimilation. Bertamini et al. [14] showed that water stress impacts grapevine leaf functioning; decreases leaves’ relative water content (RWC), dry matter, and chlorophyll content; and increases leaf proline content. Higher lipid peroxidation due to increases in hydrogen peroxide and the enhancement of antioxidant enzyme activity are common features [15].

Interactions between water stress and Botryosphaeria dieback have been investigated [1,16] by imposing water stress either before or after plant inoculation with Botryosphaeriaceae species. van Niekerk [8] evaluated the effect of water deficits in grapevines previously infected with *Neofusicoccum australe*, *Neofusicoccum parvum*, *L. theobromae*, and *Diplodia seriata*. The authors observed that the length of the lesions around the inoculated sites was higher in plants under water stress. Other experiments induced water limitations in grapevines before Botryosphaeriaceae inoculation. While *Neofusicoccum luteum* showed greater aggressiveness in plants grown under a higher soil moisture content [17], higher susceptibility to *Botryosphaeria dothidea*, *D. seriata*, *L. theobromae*, and *N. parvum* was found in grapevines under water-limiting conditions [18]. In general, results were influenced by the age and genotype of the host, type of pathogen, severity of both stressors, and the time and order of occurrence of the fungal infection and water stress. Although these reports focused on the effect of water deficits in the outcome of plant–pathogen interactions, it is difficult to ascertain the influence of stress timing in relation to infection on woody plants defence responses, or a clear role for this interaction [19]. Moreover, how abiotic stress factors may influence the transition from the endophytic to the pathogenic phases for GTD fungi is still unclear [5].

The present work aims to elucidate how water deficit timing (pre- and post-inoculation) influences disease progression in grapevines infected with *L. theobromae* by evaluating key stress-related physiological markers such as gas-exchange parameters, primary metabolism, and some hormones.

## 2. Results

### 2.1. Symptoms of Pathogen Infection

External symptoms of disease, including foliar chlorosis and wilting, varied amongst the treatments (data not shown); growth, for instance, was clearly affected by some of the treatments (Figure 1A). Grapevines with the F-WD treatment (inoculation before water deficit) exhibited the most severe disease symptoms, including wilting of shoots, foliar chlorosis, and necrosis. Furthermore, 50% of the grapevines with the F-WD treatment (inoculation after water deficit) did not survive 28 days after inoculation (Figure 1B), dictating the end of the experiment for all treatments. Internal stem lesions were visible in all inoculated grapevines, with the internal stem lesion in WD-F grapevines being half that of the F-WD (Figure 1C).

### 2.2. Plant Water Status, Leaf Gas Exchange, and Chlorophyll a Fluorescence

Plant water status was evaluated by midday stem water potential (ᴪ_md_). The water deficit alone (WD) or combined with fungal inoculation (WD-F and F-WD) induced a significant reduction in ᴪ_md_ in comparison to well-watered grapevines—the reduction being higher with the WD than with the combined stresses (Figure 2A). In spite of the slight decrease with the WD-F, non-significant differences were observed relating to water use efficiency (Figure 2B).

Leaf gas-exchange parameters varied in response to WD, fungal inoculation, and combined conditions (Figure 3). Plants under WD, WD-F, and F-WD presented similar trends for A, gs, and E—with the imposed stresses causing severe reductions (Figure 4A–C). The fungal infection in the well-watered vines (WW-F) resulted in a significant decrease in gs, and a slight decrease in A and E (Figure 3A–C). The internal CO_2_ concentration was not significantly affected (Figure 3D).

Regarding chlorophyll a fluorescence, significant decreases were observed in the ɸPSII (actual PSII efficiency) of all treatments in comparison to the control (WW; Figure 4A). A reduction in Fv/Fm (the maximum photochemical quantum efficiency of PS II) was only observed in WD-F (Figure 4B).

### 2.3. Stress-Related Metabolites: Proline and MDA

Free proline content was differentially modulated by the imposition of water stress and fungal infection (Figure 5A). Proline levels increased similarly in the WD and F-WD plants, but non-significantly in relation to WW plants. Fungal infection after water deficit (WD-F) resulted in higher proline levels in comparison to all other treatments, but only significantly compared with the WW. Regarding the MDA content, a slight, non-significant increase was observed in the WW-F, WD, and WD-F plants, while the F-WD plants exhibited a significantly lower MDA content in relation to WD-F (Figure 5B).

### 2.4. Hormonal Dynamics

In general, leaf hormonal content was influenced by the imposed water deficit and fungal infection (Figure 6). Endogenous SA content was significantly reduced in WW-F plants (Figure 6A) compared to WW. JA only showed a significant accumulation in F-WD plants, while ABA content significantly increased with all WD treatments (Figure 6C).

## 3. Discussion

This study enabled us to add new insights into abiotic and biotic stress combination in woody plants and the role of infection timing on physiological and biochemical plant responses using a relevant pathosystem as an experimental platform. Infection of grapevines with *L. theobromae* before water deficit or after a period of water deficit induced different physiological responses according to the primary stress encountered, which resulted in increased susceptibility to the pathogen or led to a primed-like state, respectively. Here, we aim to highlight the role of abiotic stress in biotic interactions—a topic usually neglected in grapevine research. Grapevines showing similar internal stem lesions had different survival rates and some different physiological responses depending on the time when the abiotic stress was imposed (pre- or post-inoculation).

Changes in the photosynthetic activity in water deficit (WD, WD-F, and F-WD) plants were revealed by: (i) lower levels of stomatal conductance (gs), (ii) decreases in CO_2_ assimilation (A), and (iii) decreases in the transpiration rate (E) compared with well-watered plants (WW, WW-F)—confirming the impact of water deficits on grapevine physiological responses [8]. The photosynthesis disturbance observed in WD grapevines seems to occur due to stomatal limitations, as observed by the low gs value [20]. The stomatal closure observed in WD in both infected and non-infected grapevines was associated with a low water potential (Figure 3A) and ABA accumulation (Figure 6C) to limit water loss [21]. Additionally, Khaleghnezhad et al. [22] showed that compared to the well-watered condition, water deficit plants showed severely decreased A, gs, and Ci, accompanied by an increase in ABA content.

The role of ABA in mediating mechanisms, whereby grapevine copes with abiotic stresses, is well reported [23]—including in water deficit scenarios [24,25]. ABA has been considered a negative regulator of disease resistance due to the interference of abscisic acid with biotic stress signalling that is regulated by SA and JA. Recent research shows that ABA could also be implicated in increasing the resistance of plants towards pathogens via its positive effect on callose deposition [26].

However, regarding gas-exchange parameters, there was no evidence of an additive effect of infection, since no significant differences were detected between WD, WD-F, and F-WD regarding A, gs, E, and ABA content. Our results show that the WD outperformed the other treatments. It should be noted that healthy leaves were chosen for the analysis of photosynthetic performance, which reinforces the absence of differences at this level. Even so, the significant declining slope of the Fv/Fm in the WD-F suggests a decrease in the photochemical efficiency potential photoinhibition that may occur when a water deficit is imposed prior to *L. theobromae* inoculation. Photoinhibition occurs when photoprotection mechanisms fail to dissipate the excess excitation energy generated under limited photosynthesis, and photo-oxidative damages lead to decreased photosynthetic efficiency and/or maximum rates [27].

Inoculation of cashew seedlings with *Lasiodiplodia theobromae* resulted in the significantly lower maximal photochemical quantum yield of PSII (Fv/Fm) compared to that for the control samples [28]. However, these studies did not include an interaction with factors such as WD. Other studies [13] reported that *g*s needed to be reduced by 60–70% before changes in the electron transport rate and non-photochemical quenching of chlorophyll fluorescence were observed [29]. Intrinsic water-use efficiency (WUEi) is also reported to decrease under more severe or long-term droughts because of damage or the inhibition of photosynthesis [30]. Even so, it was possible to identify specific characteristics of stress related to the moment of inoculation of the fungus, aligning with the idea that the interaction between hydric status and infection is quite complex and unpredictable. Studies exploring the interaction between drought and the grapevine vascular disease esca demonstrates that drought completely suppresses esca leaf symptoms, and although esca and drought both alter plant water transport and carbon balance, they do so in completely distinct ways [31]. Drought is also reported to modulate immune response resistance in grapevines challenged by Botrytis cinerea [32]. However, the relationship between mechanisms of drought tolerance and resistance to pathogens remain unknown and several actors may be involved.

Plants activate distinct defence responses depending on the life cycle of the attacker encountered. In these responses, salicylic acid (SA) and jasmonic acid (JA) play important signalling roles. Several results support the hypothesis that heat stress facilitates *L. theobromae* colonization, because of the fungus’ ability to use the phenylpropanoid precursors and SA—both compounds known to control host defence [33,34,35]. SA has been used under a water deficit due to the role of this substance in stimulating plant protection mechanisms against drought stress and the oxidative stress induced by it. Thus, its application increases the content of proline, which in addition to its protective role—along with other osmotic regulators—also improves the water status of the plant [36].

SA induces a defence against biotrophic pathogens that feed and reproduce on live host cells, whereas JA activates a defence against necrotrophic pathogens that kill host cells for nutrition and reproduction [37]. So, this could explain the highest JA concentration being obtained in the F-WD grapevines, suggesting a defence response to *L. theobromae* infection. JA synthesis is a usual feature for many plant fungal pathogen interactions or symbionts [38] including *L. theobromae*. Interestingly, JA production has only been reported in plant–fungi interactions, suggesting that these fungi may have evolved the ability to produce JA in order to colonize plants [38]. Most classical plant hormones are also produced by pathogenic and symbiotic fungi. The way in which these molecules favour the invasion of plant tissues and the development of fungi inside plant tissues is still largely unknown [39].

Plant responses to WD-F treatment seems to be mainly linked to water deficit outcomes, showing less-negative effects than F-WD in terms of survival rates. This WD-F behaviour may be because plants previously exposed to water deficits can activate a priming response which enables plants to defend from possible pathogen infection, leading to a higher survival rate than that of F-WD (Figure 1B) [40]. Barradas et al. (2018) [41] reported that water stress-primed plants were slightly more resistant to fungal infection than non-primed ones in the Eucalyptus–*Neofusicoccum eucalyptorum* pathosystem.

One of the most important responses of grapevines to WD is the overproduction of total free amino acids and proline [42,43]. The higher accumulation of proline in WD-F plants could indicate the stimulation of this defence mechanism against WD, the first threat encountered. Apart from acting as an osmolyte for osmotic adjustment, proline contributes to stabilizing sub-cellular structures (e.g., membranes and proteins), scavenging free radicals and buffering the cellular redox potential [44]. We hypothesise that this occurs because grapevines exposed to water stress followed by infection show an increase in lipid peroxidation (high concentration of), probably due to ROS production [45]. Increased membrane lipid peroxidation is proportional to the intensity of WD and is derived from the spontaneous reactions of ROS with organic molecules contained in the membranes [46]. Interestingly, WD-F showed higher MDA compared to F-WD. This could be related to the magnitude of the water deficit impact, which seems to be higher than the stress imposed by pathogen inoculation. However, we should bear in mind that in relation to WD-F and F-WD treatments, there is a gap of 14 days in water deficit treatments.

## 4. Materials and Methods

### 4.1. Plant Material

Dormant cuttings of grapevine rootstocks (1103-P) and Touriga Nacional scion (22-ISA-PT; 6 months old) were rooted in 2 L plastic pots filled with 3:2 (*w*/*w*) peat:perlite and acclimated for three months (April to July 2018) under greenhouse conditions: natural light, 60/65% relative humidity (day/night), and approximately 25 °C day/15 °C night. Plants were watered once a week until rooting and the development of the first leaves. Afterwards, all grapevines were watered up to 70% field capacity (FC) and fertigated twice a week with 5 mL/L of N:P:K nutrient solution (5:8:10, Complesal, Bayer CropScience, Carnaxide, Portugal).

Pot weight was monitored every day and the percentages of FC were maintained by adding the amount of water lost. Pots were randomly arranged throughout the experiment. Plants were left to grow with two main stems and all the growth above the ninth node was trimmed.

### 4.2. Experimental Design

The experiment was conducted between July and August 2018. Greenhouse conditions were maintained as in the acclimation period during the experiment, but the watering regime was altered, and the plants were inoculated with *L. theobromae* (isolate Bt105). Grapevine plants were randomly allocated to five different treatments (Figure 7), which included two levels of watering (water deficit and well-watered) and two levels of inoculation (inoculated and non-inoculated). To evaluate the effects of water deficit timing, two inoculation treatments were included either before or after the stress imposition. Thus, the treatments were: WW (well-watered control—70% Field Capacity); WW-F (well-watered inoculated—70% FC); WD (water deficit control 15% FC); F-WD (inoculation before water deficit–15% FC); WD-F (inoculation after water deficit 15%–FC). Each of the five treatments had five biological replicates, providing a total of 30 experimental units.

#### Water Deficit Treatment

Well-watered pots (WW and WW-F) were maintained gravimetrically every day to keep the soil water content at 70% FC by manually watering. To evaluate the effects of water deficit timing on host–fungi interactions, watering was withheld until the soil moisture content reached 15% FC, and then maintained gravimetrically daily for 14 days before inoculation (WD-F) or after inoculation (F-WS; Figure 7).

### 4.3. Fungal Culture and Plant Inoculation

The *L. theobromae* Bt105 isolate used in this study was isolated from *V. vinifera* cv. Castelão in Portugal [34]. Fungal inoculation was initiated by surface disinfection of the stems with 70% ethanol. A 5 mm area of the bark was removed with a sterile cork borer from the base of each stem between the second and third nodes. The wounds were inoculated with 5 mm mycelial plugs taken from the actively growing margin of 5-day-old colonies of *L. theobromae* growing on potato dextrose agar (PDA; VWR Chemicals, Leuven, Belgium) at 25 °C in darkness. Each inoculation point was covered with moist cotton wool and sealed with Parafilm to prevent desiccation. Plants from the WW and WD treatments were mock-inoculated with sterile 5 mm PDA plugs to assure that the observed lesions were due to infection by the pathogen and not due to wounding.

### 4.4. Survival Rate, Internal Stem Lesion, and Re-Isolation of the Pathogen

The survival rate was quantified based on the presence of living, above-ground tissues and calculated considering the initial number of plants per treatment relative to the final number of plants alive at the end of the experiment. The development of external disease symptoms (stem lesions, foliar chlorosis, and wilting) was visually assessed each week. Internal stem lesions were measured in all plants per treatment at the end of the experiment. Small pieces of necrotic tissue from the edge of each lesion were cut and placed on PDA and incubated at 25 °C in the darkness (to fulfil Koch’s postulates), followed by identification of the pathogen through micromorphological analysis.

### 4.5. Leaf Gas Exchange-Related Parameters and Plant Water Status

Net CO_2_ assimilation rate (A, μmol CO_2_ m^−2^ s^−1^), stomatal conductance (gs, mol H_2_O m^−2^ s^−1^), transpiration rate (E, mmol H_2_O m^−2^ s^−1^), and intercellular CO_2_ concentration content (Ci, vpm) were measured in five independent biological replicates per treatment using a portable infrared gas analyser (LCpro-SD, ADC BioScientific Ltd., Hoddesdon, UK) equipped with a broad-leaf chamber. To find out the saturation light intensity A/PPFD (photosynthetic photon flux density; light response curves of CO_2_ assimilation) curves were performed with the following PPFD: 2000, 1500, 1000, 750, 500, 250, 100, 50, and 0 mmol m^−2^ s^−1^. After A/PPFD data analysis, punctual measurements at saturation light intensity were performed at 1500 mmol m^−2^ s^−1^. The following conditions were maintained inside the chamber during all measurements: air flux: 200 mol s^−1^; 25 °C block temperature; and atmospheric CO_2_ and H_2_O concentration. Data were recorded when the measured parameters were stable (2–6 min). Water use efficiency (WUEi) was calculated using the formula: WUEi = A/E.

Midday stem water potential (ᴪ_md_, MPa) was measured with a Scholander-type pressure chamber (PMS Instrument Co., Albany, OR, USA) in five independent biological replicates per treatment at 12:30 (solar time), as described before [10].

### 4.6. Chlorophyll a Fluorescence Analysis

Steady-state modulated chlorophyll fluorescence was determined with a portable fluorometer (Mini-PAM; Walz, Effeltrich, Germany) on the same leaves as used for the gas-exchange measurements. Light-adapted components of chlorophyll fluorescence were measured in the midday period: steady-state fluorescence (F), maximal fluorescence (F’m), variable fluorescence F’v (equivalent to F’m − F), and quantum yield of PSII photochemistry (ɸPSII; equivalent to F’v/F’m). Leaves were then dark-adapted for at least 30 min to obtain F0 (minimum fluorescence), Fm (maximum fluorescence), Fv (variable fluorescence, equivalent to Fm − F0), and Fv/Fm (maximum quantum yield of PSII photochemistry).

### 4.7. Proline and Lipid Peroxidation Determination

Proline content was determined as described by Bates et al. [47], with slight modifications. Five leaves (50 mg) were collected per treatment, frozen in liquid N_2_, and homogenized with 750 µL sulfosalicylic acid (3%, *w*/*v*). Following centrifugation (10 min, 10,000× *g*, 4 °C), 500 µL of supernatant were collected in a new tube, and 500 µL of ninhydrin acid and 500 µL of glacial acetic acid were added. After incubation at 100 °C for 1 h and cooling on ice, 1 mL of toluene was added to the solution and the absorbance was read at 520 nm. The free proline content was calculated using a standard curve.

Lipid peroxidation was estimated by measuring the amount of malondialdehyde (MDA) in the leaves following the protocol described by Hodges et al. [48], using 50 mg of frozen leaves. Samples were extracted with 2.5 mL of 0.1% TCA (trichloroacetic acid) and vortexed. After centrifugation, an aliquot of the supernatants was added to a test tube with an equal volume of either: (1) positive (+) 0.5% (*w*/*v*) TBA solution containing 20% (*w*/*v*) TCA; or (2) negative (−) TBA solution in 20% TCA. Samples were heated at 95 °C for 30 min and, after cooling and centrifuging, the absorbance was read at 440, 532, and 600 nm. MDA equivalents (nmol mL^–1^) were calculated as (A − B/157,000) × 106, where A = [(Abs 532+TBA) − (Abs 600+TBA) − (Abs 532–TBA − Abs 600–TBA)] and B = [(Abs 440+TBA − Abs 600+TBA) × 0.0571].

### 4.8. Hormone Quantification

Salicylic acid (SA), jasmonic acid (JA), and abscisic acid (ABA) were extracted from five leaves per treatment based on Durgbanshi et al. [49]. After finely ground, freeze-dried tissue (50 mg) was mixed with internal standards (Sigma-Aldrich, Düren, Germany), 100 ng of SAd6, 100 ng of dihydrojasmonic acid, 100 ng of ABAd6, and 5 mL of distilled water. After cold centrifugation, supernatants were recovered, and the pH was adjusted to 3 with 30% acetic acid. The acidified water extract was partitioned twice against 3 mL of diethyl ether. The organic upper layer was recovered and the vacuum evaporated in a centrifuge concentrator (SpeedVac, Jouan, Saint Herblain, France). The dry residue was then resuspended in a 10% methanol solution by gentle sonication. The resulting solution was passed through 0.22 µm regenerated cellulose membrane syringe filters (Albet S.A., Barcelona, Spain) and directly injected into a UPLC system (Acquity SDS, Waters Corp., Milford, MA, USA). Analytes were separated by reverse-phase (Nucleodur C18, 1.8 µm 50 × 2.0 mm, MachereyNagel, Barcelona, Spain) using a linear gradient of ultrapure water (A) and methanol (B; both supplemented with 0.01% acetic acid) at a flow rate of 300 µL min^−1^. The gradient used was: (0–2 min) 90:10 (A:B), (2–6 min) 10:90 (A:B), and (6–7 min) 90:10 (A:B). Hormones were quantified with a Quattro LC–triple quadrupole mass spectrometer (Micromass, Manchester, UK) connected online to the output of the column through an orthogonal Z-spray electrospray ion source. The analytes were quantified after external calibration against the standards.

### 4.9. Statistical Analysis

Data are presented as mean ± standard error (SE) of five independent biological replicates. Statistical procedures were performed using SigmaPlot (Systat Software Inc., San Jose, CA, USA). After testing for ANOVA assumptions (homogeneity of variances with the Levene’s mean test, and normality with the Kolmogorov–Smirnov test, *p* ≤ 0.05), statistical differences among all treatments were evaluated by one-way analysis of variance (ANOVA, *p* ≤ 0.05) followed by post-hoc multiple comparisons using the Tukey test. Different lower cases indicate significant differences among the treatments (WW, WW-F, WD, F-WD, and WD-F) at *p* ≤ 0.05.

## 5. Conclusions

There have been very few studies experimentally addressing the effects of the timing of drought on physiological responses of woody plants to pathogen infection [19,50]. With this work, we hope to have contributed to this gap in knowledge by addressing a biological system that is extremely important economically but still subject to many challenges.

Our results illustrate different specific responses in the interactions between biotic and abiotic stress in grapevines that are dependent on the order of stress imposition. The percentage of survival decreased only when WD was imposed after *L. theobromae* inoculation, contrasting with the priming effect verified when WD was imposed prior to fungal inoculation. Overall, grapevines respond differently to the primary stress encountered, with consequences at different physiological and—mainly—biochemical responses, which were not found in internal stem lesions, but were clear in the grapevines’ survival rate.

## Figures and Tables

**Figure 1 plants-11-01961-f001:**
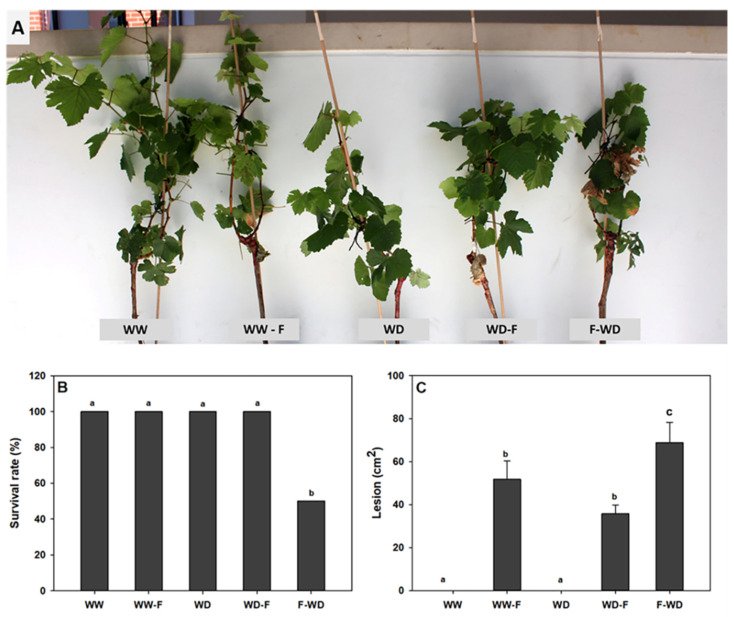
(**A**) External disease symptoms, (**B**) survival rate (%), and (**C**) internal stem lesion (cm^2^) of grapevines exposed to WW (well-watered control); WW-F (well-watered fungal inoculation); WD (water deficit); F-WD (inoculation before water deficit); WD-F (inoculation after water deficit). Different lowercase letters indicate significant differences among the five treated groups (*p* ≤ 0.05).

**Figure 2 plants-11-01961-f002:**
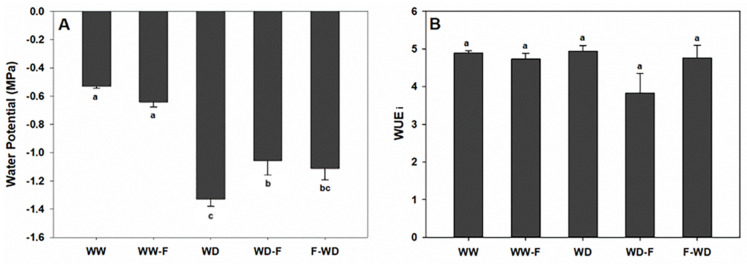
Midday stem water potential ((**A**): ᴪ_md_) and water use efficiency (**B**) of grapevines exposed to WW (well-watered control); WW-F (well-watered fungal inoculation); WD (water deficit); F-WD (inoculation before water deficit); WD-F (inoculation after water deficit). Different lowercase letters indicate significant differences among the five treated groups (*p* ≤ 0.05).

**Figure 3 plants-11-01961-f003:**
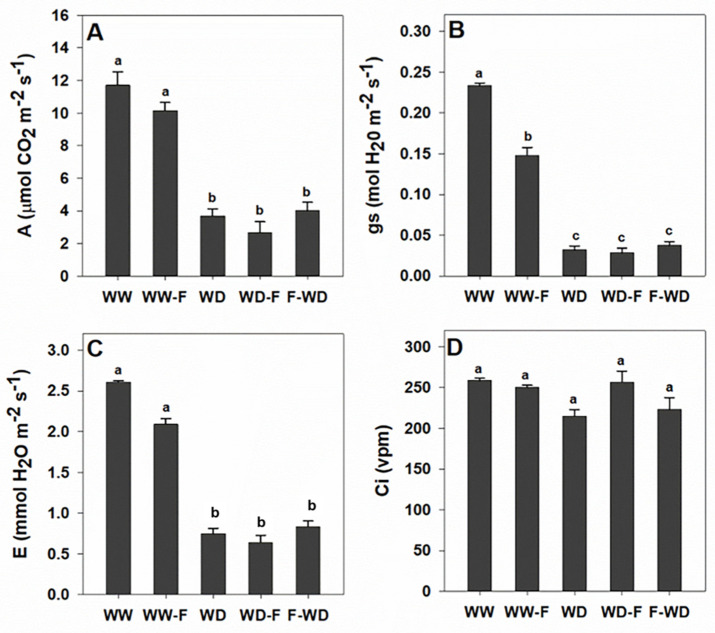
(**A**) Net photosynthetic rate (A), (**B**) stomatal conductance (gs), (**C**) transpiration rate (E), and (**D**) internal CO2 concentration (Ci) of grapevine plants exposed to WW (well-watered control); WW-F (well-watered fungal inoculation); WD (water deficit); F-WD (inoculation before water deficit); WD-F (inoculation after water deficit). Different lowercase letters indicate significant differences among the five treated groups (*p* ≤ 0.05).

**Figure 4 plants-11-01961-f004:**
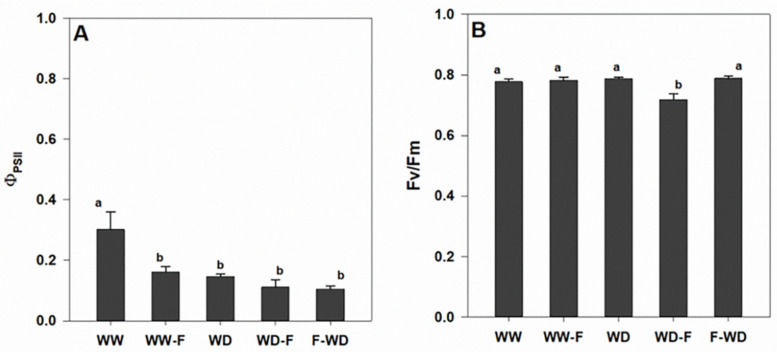
(**A**) Midday quantum yield of PSII photochemistry (ΦPSII) and (**B**) maximum quantum yield of PSII photochemistry (Fv/Fm) of grapevines exposed to WW (well-watered control); WW-F (well-watered fungal inoculation); WD (water deficit); F-WD (inoculation before water deficit); WD-F (inoculation after water deficit). Different lowercase letters indicate significant differences among the five treated groups (*p* ≤ 0.05).

**Figure 5 plants-11-01961-f005:**
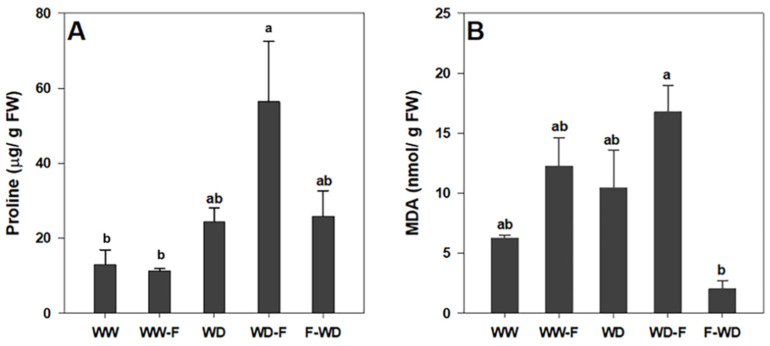
(**A**) Proline content and (**B**) malondialdehyde content (MDA) of grapevines exposed to WW (well-watered control); WW-F (well-watered fungal inoculation); WD (water deficit); F-WD (inoculation before water deficit); WD-F (inoculation after water deficit). Different lowercase letters indicate significant differences among the five treated groups (*p* ≤ 0.05).

**Figure 6 plants-11-01961-f006:**
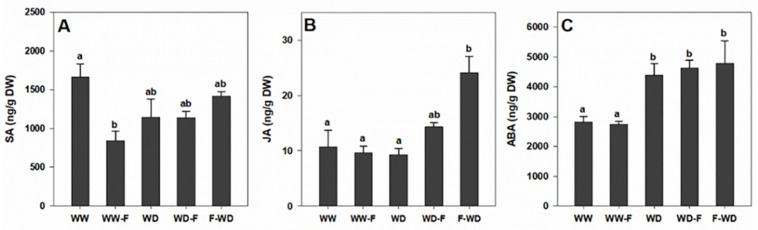
(**A**) Salicylic acid (SA), (**B**) jasmonic acid (JA), and (**C**) abscisic acid (ABA) of grapevines exposed to WW (well-watered control); WW-F (well-watered fungal inoculation); WD (water deficit); F-WD (inoculation before water deficit); WD-F (inoculation after water deficit). Different lowercase letters indicate significant differences among the five treated groups (*p* ≤ 0.05).

**Figure 7 plants-11-01961-f007:**
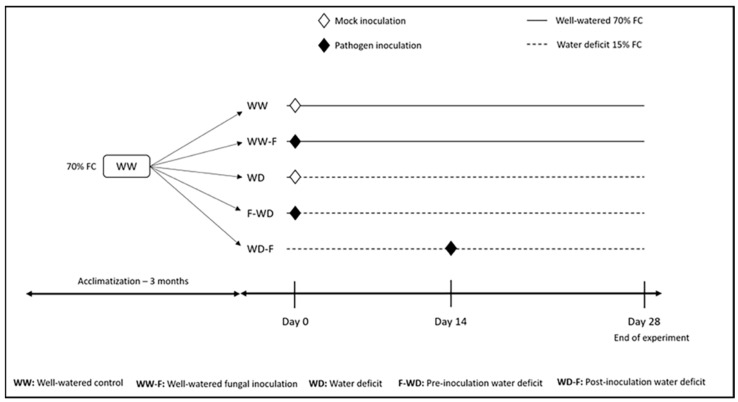
Schematic illustration of the experimental design.

## Data Availability

Not applicable.

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
