# Peer review of "Water Deficit Timing Differentially Affects Physiological Responses of Grapevines Infected with Lasiodiplodia theobromae"

_plants, 2022, doi:10.3390/plants11151961_

Round 1

Reviewer 1 Report

The manuscript reports on the combined effect of water deficit and fungal infection in young grapevines. The latter, with inoculation before and after the imposition of water deficit. Observations include visual symptoms, gas exchange and photosynthesis, some metabolites related to stress and plant hormones.

The methods seem appropriate, but it would be convenient to add some details, s

In general, the manuscript is well written, shows a relevant subject in grapevine physiology regarding the interaction of two recurrent stresses: water deficit and wood fungal infections.

1.     It is not clear to me however, the relevance of some data. Ci, for instance does not seem to be considered in discussion (and just a bit in results). Nor the WUE. In fact, I would suggest eliminating Ci and, instead, to add the iWUE in its place in figure 3 (since iWUE is derived from AN and E and is not related to water potentials).

2.     Another point is the discussion of the photosynthesis data. It is not completely clear, from the discussion section, why, for instance, the Fv/Fm is reduced in WD-F, when compared to WD and F-WD there are not differences in gs nor in AN or FPSII. Would this mean that the Fv/Fm decline is not exclusively driven by excess energy but some other signals/factors? I think the discussion should make comments on it. Regarding this, the text says, “The significant declining slope of Fv/Fm in WD-F suggests a decrease in photochemical efficiency potential photoinhibition that may occur when water deficit is imposed prior to L. theobromae inoculation”. But that is an observation, not a discussion of the fact, really.

3.     Finally, and I do not have a strong background in phytopathology, I miss in the discussion the relation of the fungal infection on stem xylem vessels and the changes in plant hormones at the leaf level. In relation to the following sentence in the text (Line 192): “SA induces defense against biotrophic pathogens that feed and reproduce on live host cells, whereas JA activates defense against necrotrophic pathogens that kill host cells for nutrition and reproduction”, it is mentioned tissues that are not involved in the infection. Xylem vessels are dead cells, again, away from the leaves. Could the authors further develop the relationship between fungal infection and the hormone responses at the leaf level, in the discussion section?

Line 37: right bracket missing.

Line 90: I am not sure that chlorosis is clear from figure 1, so I would state “data not shown”, or to make comments in relation to figure 1 (growth, for instance is clearly affected by some of the treatments)

Line 93-94: I do not see in figure 7 a support for the previous sentence.

Line 103: instead of á´ªmd the “md” should be in lower script

Line 109 (and else) WUE should be WUEi (Intrinsic, es derived from AN/E).

In M&M, Line 291 should be MPa

Author Response

Reviewer #1

  1. It is not clear to me however, the relevance of some data. Ci, for instance does not seem to be considered in discussion (and just a bit in results). Nor the WUE. In fact, I would suggest eliminating Ci and, instead, to add the iWUE in its place in figure 3 (since iWUE is derived from AN and E and is not related to water potentials).

Reply: We are thankful for the reviewer’s comments however, we decided to keep the graph referring to Ci as it ends up indirectly supporting some of the results that are now discussed. The fact that there are no differences reinforces potential biochemical problems that limit photosynthesis beyond stomatal limitation depending on the treatments imposed. This information is discussed and the presence of these results is important to validate the idea.

  1. Another point is the discussion of the photosynthesis data. It is not completely clear, from the discussion section, why, for instance, the Fv/Fm is reduced in WD-F, when compared to WD and F-WD there are not differences in gs nor in AN or FPSII. Would this mean that the Fv/Fm decline is not exclusively driven by excess energy but some other signals/factors? I think the discussion should make comments on it. Regarding this, the text says, “The significant declining slope of Fv/Fm in WD-F suggests a decrease in photochemical efficiency potential photoinhibition that may occur when water deficit is imposed prior to theobromae inoculation”. But that is an observation, not a discussion of the fact, really.

Reply: We understand the doubt raised by the reviewer and have tried to improve the discussion in this section. The WD-F treatment gives rise to a combined stress that is actually a new stress and cannot be compared with either the isolated stress or the F-WD stress sequence. WD treatments impair CO2 fixation at a level that the rate of reduction of energy use by the Calvin cycle is lower than the rate of its production. Probably for WD-F treatments this lead to the need for more protective mechanisms against excess reducing energy than other treatments. These photoprotective mechanisms compete with photochemistry for absorbed energy, leading to a decrease in the quantum yield of PSII.

  1. Finally, and I do not have a strong background in phytopathology, I miss in the discussion the relation of the fungal infection on stem xylem vessels and the changes in plant hormones at the leaf level. In relation to the following sentence in the text (Line 192): “SA induces defense against biotrophic pathogens that feed and reproduce on live host cells, whereas JA activates defense against necrotrophic pathogens that kill host cells for nutrition and reproduction”, it is mentioned tissues that are not involved in the infection. Xylem vessels are dead cells, again, away from the leaves. Could the authors further develop the relationship between fungal infection and the hormone responses at the leaf level, in the discussion section?

Reply: In this work we are assessing the impact of infection on the plant response. Our aim is not to see what happens at the site of infection but rather how the plant copes with it and how this alters key processes related to primary and secondary metabolism and the role of signalling agents such as hormones. This is the most conventional approach in plant-pathogen studies and is essential for identifying phenotypic traits that could be used as markers of resistant plant selection or innovative disease control strategies. Understanding the interaction between biotic and abiotic factors is complex because of all the variables that can be equated. These studies are still in their infancy for grapevine but are essential to understand more realistically the response of plants. We believe that our work contributes to this gap in knowledge and opens up new questions to be considered in the future. This section has been improved

Line 37: right bracket missing.

Reply: Is now revised

Line 90: I am not sure that chlorosis is clear from figure 1, so I would state “data not shown”, or to make comments in relation to figure 1 (growth, for instance is clearly affected by some of the treatments).

Reply: We are revised and followed the reviewer’s comments.

Line 93-94: I do not see in figure 7 a support for the previous sentence.

Reply: We deleted the reference to the figure 7.

Line 103: instead of á´ªmd the “md” should be in lower script

Reply: The “md” is now in lower script.

Line 109 (and else) WUE should be WUEi (Intrinsic, es derived from AN/E).

Reply: The WUE was replaced to WUEi.

In M&M, Line 291 should be MPa

Reply: corrected  

Reviewer 2 Report

The authors report an interesting analysis of the response of Vitis vinifera to abiotic and biotic stresses. In particular, the authors defined different V. vinifera responses depending on whether the infection of the fungus, Lasiodiplodia theobromae, occurs before or after water stress. The results are very encouraging. In fact, V. vinifera, if subjected to water stress, responds better to the attack of the fungus in terms of survival and the extent of leaf lesions. Interesting data are observed on the parameters of proline and MDA production.

Critical points:

1) However, the physiological parameters on photosynthesis are not particularly different between plants that receive the fungus before or after water stress. This aspect should be clarified and discussed better.

2) If the authors quantified hormones with analytical techniques it would be important to include chromatographic profiles;

3) Also with regard to the hormonal signals of SA, ABA, and JA, no considerable differences are observed. This aspect should be better discussed. I suggest you see and include some recent publications on this:

Concentrations-dependent effect of exogenous abscisic acid on photosynthesis, growth and phenolic content of Dracocephalum moldavica L. under drought stress; 10.1007/s00425-021-03648-7;

Yousefvand, P.; et al. Salicylic Acid Stimulates Defense Systems in Allium hirtifolium Grown under Water Deficit Stress. Molecules 2022, 27, 3083. doi: 10.3390/molecules27103083

Author Response

Reviewer #2

The authors report an interesting analysis of the response of Vitis vinifera to abiotic and biotic stresses. In particular, the authors defined different V. vinifera responses depending on whether the infection of the fungus, Lasiodiplodia theobromae, occurs before or after water stress. The results are very encouraging. In fact, V. vinifera, if subjected to water stress, responds better to the attack of the fungus in terms of survival and the extent of leaf lesions. Interesting data are observed on the parameters of proline and MDA production.

Reply: We are thankful for the reviewer’s comments.

1 - However, the physiological parameters on photosynthesis are not particularly different between plants that receive the fungus before or after water stress. This aspect should be clarified and discussed better.

Reply: Indeed, the responses at the photosynthetic level are not very different and are more related to the WD effect. It should be noted that the leaves selected for analysis were, even for the treatment in which plants died, healthy, without necrosis and therefore it would not be expected that there would be many differences in the inoculation effect at this level (with the exception of the photochemical performance for the WD-F treatment). Still, it was possible to identify specificities in important signalling agents in plant physiological processes for the WD-F and F-WD treatments. This work indicates new lines of investigation such as exploring secondary metabolism to better understand plant-plant interactions.  The discussion was improved.

2 - If the authors quantified hormones with analytical techniques it would be important to include chromatographic profiles;

Reply: The hormonal quantification was performed as in Durgbanshi et al (2005) J. Agric. Food Chem. 53, 8437-8442. In that paper, all chromatographic and mass spectrometry procedures are detailed. Providing raw data in a regular article seems an unconventional practice that does not provide much information. However, if needed we could provide some examples of chromatographic profiles in the supplementary material.

3 - Also with regard to the hormonal signals of SA, ABA, and JA, no considerable differences are observed. This aspect should be better discussed. I suggest you see and include some recent publications on this:

  • Concentrations-dependent effect of exogenous abscisic acid on photosynthesis, growth and phenolic content of Dracocephalum moldavica L. under drought stress; 10.1007/s00425-021-03648-7;
  • Yousefvand, P.; et al. Salicylic Acid Stimulates Defense Systems in Allium hirtifolium Grown under Water Deficit Stress. Molecules 2022, 27, 3083. doi: 10.3390/molecules27103083

Reply: We do not understand the reviewer's comment as we clearly have ABA level responses in the WD treatments and in JA for the F-WD treatment. SA also decreases in all treatments that include a stressor. We agree that this topic could be improved and that is what we have done.  We have also included references that we think are relevant to support some ideas.

Reviewer 3 Report

Dear authors,

you present a study on the effect and timing of two stress factor (drought (WD), fungal infection(F)) on the response of grapevines. You hypothesize that WD prior to F could prime stress reactions to the fungal infection. You clearly show that disease severity is more pronounced when fungal infections occur prior WD. And that fungal infections after WD show same disease incidence as well watered plants. To explain this effect you present data on water status, photosynthesis, stress metabolites, and hormones.

All parts of the manuscript are writte in a concise manner, giving all relevant information needed to understand the study. I only have made a few comments in the attached PDF file.

What I would suggest to highlight your main findings in the discussion much more or to add "conclusions" which I miss. After reading your article the major findings are (for me)

(1) WD before F is able to keep disease on the level of WW-F plants. It seems important for me to stress this point, because it is indicating a priming of WD.

(2) Water status is mainly driven by WD (fig. 2), which is a bit surprising for me that a fungal infection will not increase the effect.

(3) Photosynthesis is mainly driven by WD (fig. 3), but photosynthetic aparatus is equally affected by all stress types (fig. 4)

(4) Proline is increased highest in the WD-F combination, while MDA is reduced greates in F-WD (fg. 5)

(5) hormones show a specific pattern. SA is greatest reduced in WW-F (fungal infection is driving factor); JA is mainly driven by a F-WD combination; ABA is driven by drought (fig. 6)

You have addressed these findings briefly in the abstract. I would give these key findings more room in a special paragraph. Especially, because in the discussion some of these points could easily be overseen. I would try to highlight these 5 key results in a conclusion. I strongly suggest to add a conclusion!

Author Response

Reviewer #3

You present a study on the effect and timing of two stress factor (drought (WD), fungal infection(F)) on the response of grapevines. You hypothesize that WD prior to F could prime stress reactions to the fungal infection. You clearly show that disease severity is more pronounced when fungal infections occur prior WD. And that fungal infections after WD show same disease incidence as well watered plants. To explain this effect you present data on water status, photosynthesis, stress metabolites, and hormones. All parts of the manuscript are writte in a concise manner, giving all relevant information needed to understand the study. I only have made a few comments in the attached PDF file.

What I would suggest to highlight your main findings in the discussion much more or to add "conclusions" which I miss. After reading your article the major findings are (for me)

(1) WD before F is able to keep disease on the level of WW-F plants. It seems important for me to stress this point, because it is indicating a priming of WD.

(2) Water status is mainly driven by WD (fig. 2), which is a bit surprising for me that a fungal infection will not increase the effect.

(3) Photosynthesis is mainly driven by WD (fig. 3), but photosynthetic aparatus is equally affected by all stress types (fig. 4)

(4) Proline is increased highest in the WD-F combination, while MDA is reduced greates in F-WD (fg. 5)

(5) hormones show a specific pattern. SA is greatest reduced in WW-F (fungal infection is driving factor); JA is mainly driven by a F-WD combination; ABA is driven by drought (fig. 6)

You have addressed these findings briefly in the abstract. I would give these key findings more room in a special paragraph. Especially, because in the discussion some of these points could easily be overseen. I would try to highlight these 5 key results in a conclusion. I strongly suggest to add a conclusion!

Reply: We are thankful for the reviewer’s comments which in general contributed to improve the manuscript. The discussion has been improved to highlight some of the aspects pointed out and there is a section dedicated to conclusions.

Round 2

Reviewer 2 Report

The authors have only partially improved the manuscript. I believe that the authors should better discuss some aspects of the hormonal signal by commenting on the suggested references:

  • Concentrations-dependent effect of exogenous abscisic acid on photosynthesis, growth and phenolic content of Dracocephalum moldavica L. under drought stress; 10.1007/s00425-021-03648-7;
  • Yousefvand, P.; et al. Salicylic Acid Stimulates Defense Systems in Allium hirtifolium Grown under Water Deficit Stress. Molecules 2022, 27, 3083. doi: 10.3390/molecules27103083

Author Response

Dear reviewer, we agree that this topic could be improved and we have did it. We have also included the suggested references to support some ideas.

Round 3

Reviewer 2 Report

The authors have greatly improved the manuscript, it can now be published